# Metabolic-Associated Fatty Liver Disease and Insulin Resistance: A Review of Complex Interlinks

**DOI:** 10.3390/metabo13060757

**Published:** 2023-06-15

**Authors:** Thomas M. Barber, Stefan Kabisch, Andreas F. H. Pfeiffer, Martin O. Weickert

**Affiliations:** 1Warwickshire Institute for the Study of Diabetes, Endocrinology and Metabolism, University Hospitals Coventry and Warwickshire, Clifford Bridge Road, Coventry CV2 2DX, UK; 2Division of Biomedical Sciences, Warwick Medical School, University of Warwick, Coventry CV4 7AL, UK; 3NIHR CRF Human Metabolism Research Unit, University Hospitals Coventry and Warwickshire, Clifford Bridge Road, Coventry CV2 2DX, UK; 4Department of Endocrinology and Metabolic Medicine, Campus Benjamin Franklin, Charité University Medicine, Hindenburgdamm 30, 12203 Berlin, Germany; 5Deutsches Zentrum für Diabetesforschung e.V., Geschäftsstelle am Helmholtz-Zentrum München, Ingolstädter Landstraße, 85764 Neuherberg, Germany; 6Centre for Sport, Exercise and Life Sciences, Faculty of Health & Life Sciences, Coventry University, Coventry CV1 5FB, UK

**Keywords:** metabolic-associated fatty liver disease, insulin resistance

## Abstract

Metabolic-associated fatty liver disease (MAFLD) has now surpassed alcohol excess as the most common cause of chronic liver disease globally, affecting one in four people. Given its prevalence, MAFLD is an important cause of cirrhosis, even though only a small proportion of patients with MAFLD ultimately progress to cirrhosis. MAFLD suffers as a clinical entity due to its insidious and often asymptomatic onset, lack of an accurate and reliable non-invasive diagnostic test, and lack of a bespoke therapy that has been designed and approved for use specifically in MAFLD. MAFLD sits at a crossroads between the gut and the periphery. The development of MAFLD (including activation of the inflammatory cascade) is influenced by gut-related factors that include the gut microbiota and intactness of the gut mucosal wall. The gut microbiota may interact directly with the liver parenchyma (through translocation via the portal vein), or indirectly through the release of metabolic metabolites that include secondary bile acids, trimethylamine, and short-chain fatty acids (such as propionate and acetate). In turn, the liver mediates the metabolic status of peripheral tissues (including insulin sensitivity) through a complex interplay of hepatokines, liver-secreted metabolites, and liver-derived micro RNAs. As such, the liver plays a key central role in influencing overall metabolic status. In this concise review, we provide an overview of the complex mechanisms whereby MAFLD influences the development of insulin resistance within the periphery, and gut-related factors impact on the development of MAFLD. We also discuss lifestyle strategies for optimising metabolic liver health.

## 1. Introduction

Globally, obesity prevalence has tripled over the last 50 years, with obesity and overweight now affecting 650 million and 1.9 billion people, respectively [1]. Obesity has a substantial impact on work productivity [2], psycho-social functioning [3], and global healthcare expenditure [4]. Furthermore, ‘obesity-associated metabolic dysfunction’ (OAMD) underlies much of obesity-associated co-morbidity and premature mortality, as evidenced by data from the Framingham Heart Study [5]. OAMD is an umbrella term that incorporates, most notably, type 2 diabetes mellitus (T2D), the cardinal features of the metabolic syndrome (dysglycaemia, dyslipidaemia, and hypertension), polycystic ovary syndrome (PCOS), obstructive sleep apnoea (OSA), and multiple malignancies [6]. The pathophysiology of OAMD stems from dysregulation of the insulin signalling pathway (insulin resistance [IR]) that is, in turn, driven by weight-gain and obesity. As such, IR, hyperinsulinaemia and its associated chronic inflammatory milieu and oxidative stress underlies much obesity-related chronic disease [7]. Multiple complex pathways mediate the effects of weight gain and obesity on the development of IR (detailed elsewhere). Within this milieu, the liver plays a key role, and acts as a linchpin mediating metabolic signals between the gut (including dietary, microbiota and complex endocrine and autonomic signaling) and the rest of the body.

Within the realm of OAMD, metabolic-associated fatty liver disease (MAFLD) lies centre stage [8], and now represents the most common cause of chronic liver disease (global prevalence ≥ 25%), and an important cause of cirrhosis and hepatocellular carcinoma [9,10]. (In this review, we use the term ‘MAFLD’ rather than ‘non-alcoholic fatty liver disease’ [NAFLD] to accurately reflect the underlying pathophysiology and metabolic implications of MAFLD, including the mechanisms underlying IR. However, the two terms, MAFLD and NAFLD, can be used interchangeably). MAFLD is typified by an accumulation of triglycerides within hepatocytes in non-alcohol users [11]. In around 10% of patients, MAFLD can progress to non-alcoholic steatohepatitis (NASH), mediated by complex lipotoxic pathways that implicate endoplasmic reticulum stress, and lysosomal and mitochondrial dysfunction [12]. The levels of lipid within hepatocytes are regulated by the interplay between the hepatic delivery and uptake of lipids, and their synthesis from carbohydrates, cleavage, metabolization to ketone bodies, ß-oxidation and secretion within very low-density lipoproteins (VLDLs) [13]. MAFLD emerges from a complex interplay between background genetic susceptibility and multiple environmental factors that include physical fitness, lifestyle, diet, sleep sufficiency, and stress [14]. The development of MAFLD is also likely influenced by gut microbiota.

The liver sits at a crossroads between the gut and the periphery. Complex interactions manifest between the gut and the liver (including the status of the gut microbiota and intactness of the gut wall), and between the liver and peripheral tissues. Such interactions are mediated primarily via the portal vein and systemic circulation, respectively. The liver mediates the metabolic status of peripheral tissues (including insulin sensitivity) through a complex interplay of hepatokines, liver-secreted metabolites, and liver-derived micro (mi)RNAs. Furthermore, gut-related factors represent an important determinant of liver health. Therefore, rather than being an ‘innocent bystander’, the liver plays a key central role in influencing overall metabolic status.

The importance of the healthiness of the gut in determining metabolic liver health provides a key to optimising metabolic health (including as a treatment and preventive approach to MAFLD) that implicates the improvement of the overall healthiness of the gut microbiota and gut wall. The healthiness of our gut reflects our lifestyle behaviours. Optimisation of gut health requires the adoption of a healthy lifestyle. This includes the avoidance of smoking and excessive weight gain (and effective weight loss in the context of obesity), reduced stress, regular exercise, healthy diet, and importantly, sleep sufficiency (including a regular sleep–wake cycle to entrain a healthy circadian rhythm within the gut microbiota) [15]. Regarding a healthy diet, fibre represents a key macronutrient known to be deficient in populations from Europe and the US [16]. Intake of dietary fibre improves insulin sensitivity [17,18,19,20,21,22]. Our own group published data from the ProFiMet cohort (overweight adults [n = 111] with features of the metabolic syndrome), in which those assigned to a high cereal fibre diet (over 18 weeks) had a significant improvement in insulin sensitivity compared to the group assigned to an isoenergetic high protein diet [23]. It is likely that the high fibre content of high-fibre cereals improves the overall healthiness of the gut microbiota and intactness of the gut wall, and that these changes facilitate improved metabolic liver health, that in turn mediates improved overall metabolic health (including insulin sensitivity).

In this concise review (summarized in Figure 1), we provide an overview of the complex interlinks between MAFLD and the development of insulin resistance. We outline the multiple pathways whereby the liver, in the context of MAFLD, mediates metabolic effects on insulin signaling within the periphery, including through the secretion of hepatokines and metabolites, and miRNAs. We provide an overview of the metabolic interplay between gut-related factors and the liver in the context of MAFLD. We conclude with a focus on optimised metabolic liver health as a sensible strategy for the future management of OAMD.

## 2. Association between MAFLD and IR

Epidemiologically, there is a close association between hepatic steatosis and IR [13,24]. Furthermore, hepatic steatosis (independent of adiposity), is associated with impaired insulin action within the liver, but also within the periphery (including adipose tissue and skeletal muscle), in both lean and non-diabetic obese people [13,25]. Relatively small increases in liver fat are associated with worsening IR within the liver and skeletal muscle [13,26]. Accordingly, far from being a benign condition, hepatic steatosis is a harbinger of metabolic disorders (particularly in the lean population), mediated through changes in endocrine and paracrine functions to cause IR in key tissues implicated in glucose and lipid homeostasis [13,27].

Although hepatic steatosis predicts the onset of IR and metabolic dysfunction, it is important to note that IR also predicts the development of MAFLD [13]. Such a scenario stems from impaired suppression of lipolysis with enhanced delivery of free fatty acids to the liver, combined with increased de novo lipogenesis [13,28]. Interestingly, although a clear association exists between MAFLD-related hepatic steatosis and IR, some genetically determined forms of fatty liver (such as a sequence variation within the ‘patatin-like phospholipase domain-containing protein 3′ [*PNPLA3*] gene) do not associate with IR [13,29]. Such observations provide insights into possible mechanistic links between MAFLD and IR, and novel therapeutic opportunities [13].

In the context of IR, chronic hyperinsulinaemia may contribute toward the worsening of fat accumulation within the liver through forkhead transcription factors [30,31]. Insulin activates the insulin receptor substrates, IRS1 and IRS2, with consequent inhibition of the forkhead transcription factors, FOXO1 and FOXA2 [30]. The IRS2 pathway inhibits FOXO1 expression, which results in the suppression of hepatic gluconeogenesis. Conversely, inhibition of FOXA2 results in the suppression of hepatic fatty acid oxidation [30]. Importantly, FOXA2 expression may be regulated by insulin-responsive pathways independent of IRS1 and IRS2 and, therefore, may maintain relative insulin sensitivity compared with the regulation of FOXO1 expression in the context of IR [30]. Accordingly, in insulin-resistant states, insulin fails to inhibit the hepatic production of glucose, thereby enhancing hyperinsulinaemia. Concurrently, through the relative sparing of suppressed fatty acid oxidation, triglycerides accumulate within the liver resulting in hepatic steatosis [30]. Furthermore, activation of the lipogenic enzymes, Sterol Regulatory Element-Binding Protein 1 (SREBP-1c) and Carbohydrate Response Element Binding Protein (ChREBP) within the liver in the context of hyperglycaemia and hyperinsulinaemia, contributes toward the worsening of the accumulation of triglycerides within the liver [30]. Therefore, the pathogenesis of MAFLD in the context of IR can be viewed as a vicious circle typified by dysglycaemia, hyperinsulinaemia, intra-hepatic triglyceride accumulation, and worsening hepatic IR.

There is a close association between MAFLD and T2D, with frequent co-occurrence of the two conditions [32,33]. Furthermore, it is estimated that MAFLD approximately doubles the risk of developing T2D as an independent risk factor, this risk paralleling the severity of MAFLD [32]. Improvement or resolution of MAFLD also reduces the risk of T2D, providing a rationale for liver-focused pharmacotherapies to improve overall metabolic status [32]. An important consideration is that the pathogenesis of T2D is typified by a progressive β-cell decline in insulin release that would diminish any hyperinsulinaemic effects [34]. There are complex interlinks between MAFLD, peripheral IR, and T2D. It seems likely that when MAFLD and T2D co-exist, the role of hyperinsulinaemia in the pathogenesis of hepatic steatosis and MAFLD development, as outlined above, would have a greater impact in the early stages of T2D development when β-cell insulin release is still relatively preserved.

## 3. Effects of MAFLD on Peripheral IR

The close epidemiological association between MAFLD, IR, and related impairments of glucose and lipid control requires an explanation. In this section, we provide an overview of the multiple liver-derived factors that likely mediate causally the complex association between MAFLD and peripheral IR (discussed in detail elsewhere [35]). This discussion includes the role of hepatokines, liver-secreted metabolites, and liver-derived miRNAs (summarized in Table 1).

### 3.1. Hepatokines

Based on data from mass spectrometry proteomics, it is likely that the human liver encodes > 4000 secreted proteins [13,36,37]. Quantitatively, the hepatocyte is the most important cell type implicated in liver protein secretion [13,38]. The liver is ideally placed anatomically and structurally (receiving around 25% of cardiac output) to communicate with the periphery through the release of hepatokines [13]. Broadly, the proteins that form hepatokines are sub-divided into ‘classical’ (synthesized through the organelles of the secretory pathway, including the endoplasmic reticulum and Golgi, and contain a signal peptide on the N-terminus) and ‘non-classical’ (proteins secreted as ‘cargo’ within extracellular vesicles that do not contain an N-terminus signal peptide) [13]. Data from rodent-based studies (comparison of quantitative proteomic data from healthy and steatotic livers) reveal that protein signals that originate from the steatotic liver induce IR and inflammatory pathways within cells from the periphery [13]. The topic of heptatokines and MAFLD has been reviewed in detail elsewhere [13]. Here, we provide an overview of data for some selected hepatokines.

Activin E, a member of the Transforming Growth Factor β (TGFβ) family, is elevated in the serum and liver in humans with both obesity [13,39] and MAFLD [13]. However, in rodent-based studies, activin E appears to drive uncoupled respiration and prevent weight gain [13].

Fetuin-A is a glycoprotein secreted by the liver [13]. Its blood level correlates with circulating levels of triglycerides and the severity of MAFLD [13,40] and IR [13,41]. Fetuin-A provides a mediating pathway between MAFLD and IR through direct effects on the insulin receptor pathway [13]. This includes inhibition of insulin receptor tyrosine kinase activity and lower autophosphorylation [13,42]. Fetuin-A is also an endogenous ligand for toll-like receptor 4 (TLR4), enabling saturated free fatty acids to induce insulin resistance through activation of TLR4 signaling [13,43]. Fetuin-A impairs glucose sensing within the β-cells and impairs insulin secretion in response to inflammatory processes [13,44]. Finally, Fetuin-A induces adipokines such as tumour necrosis factor-α (TNF-α) and Interleukin-6 (IL-6) that, in turn, may modulate hepatocellular signaling pathways [35]. Such bi-directional hepatocyte–adipocyte crosstalk likely contributes towards the mediation of peripheral IR in MAFLD.

Fetuin B, encoded by the *FETUB* gene, is increased in the serum in patients with MAFLD and T2D and correlates with IR [13,40]. Data from human- and rodent-based studies suggest that fetuin B suppresses the ability of glucose to promote its own disposal (independent of insulin), and also impairs first-phase glucose-stimulated insulin secretion [13]. Rodent-based studies on obese, insulin-resistant mice in which Fetuin B secretion from the liver is suppressed confirm its role in MAFLD-induced IR [13,45].

Visfatin functions as an intracellular enzyme (nicotinamide phosphoribosyltransferase [NAMPT]) that mediates the synthesis of nicotinamide adenine dinucleotide, and as a cytokine-like soluble factor that is secreted into the extracellular space [46]. The former is implicated in cellular metabolism and mitochondrial biogenesis, and the latter in the induction of pro-inflammatory cytokine production [46]. Evidence from rodent-based studies [47] and in vitro assessment of hepatocytes [48] support a role for NAMPT in the development of hepatic steatosis, inflammation, and fibrosis [46]. The current literature reveals opposing activities of NAMPT in the development of MAFLD, possibly from the different roles of this enzyme in its intra- and extracellular locations [46].

Chemerin is a chemotactic adipokine secreted as an inactive precursor that is then activated by proteases [46]. Chemerin binds to the G protein–coupled receptor, chemokine-like receptor 1 (CMKLR1), expressed in macrophages, dendritic cells, and natural killer cells, and thereby may play a role in regulating immunity [46]. Chemerin release and expression of its receptor CMKLR1 are both increased in obesity and insulin-resistant states and expression of each decreases following weight loss [46]. Furthermore, chemerin may regulate glucose and lipid homeostasis, insulin sensitivity, and adipocyte differentiation [46]. However, the role of chemerin in MAFLD remains unclear, with inconsistencies in the reported data from rodent-based models [46,49,50].

Dipeptidyl peptidase-4 (DPP4), a serine protease, is secreted by the liver and rapidly inactivates the circulating incretin hormones, gastric inhibitory peptide (GIP), and glucagon-like peptide 1 (GLP1) [13]. GIP and GLP1 promote insulin secretion from the β-cell and suppress glucagon secretion from the α-cell in the post-prandial phase, thereby maintaining euglycaemia following a meal (including hepatic glucose output and enhanced peripheral glucose uptake) [13]. Human-based studies on MAFLD in the context of IR reveal elevated plasma DPP4 activity, and lower levels of serum GIP and GLP1 [13,51,52]. Rodent-based studies confirm a role for DPP4 in impaired whole-body glucose tolerance and reduced circulating GLP1 [13,51]. DPP4 is known to promote hepatic steatosis through increased fatty acid uptake into hepatocytes [13,53]. Furthermore, the clinical usage of DPP4 inhibitor therapies reveal improved hepatic steatosis and glucose tolerance [13]. It should be noted, though, that the effects of DPP4 inhibitor therapies on hepatic steatosis (including inflammatory markers and fibrosis in MAFLD) are relatively small and questionable [54,55]. Although a receptor for DPP4 has not yet been identified [13], these data do suggest direct autocrine and paracrine roles for DPP4 within the liver, in addition to peripheral effects on glucose handling through its systemic actions on the incretin hormones, GIP and GLP1.

Fibroblast growth factor 21 (FGF21) is implicated in the regulation of systemic lipid metabolism in response to diet, exercise, and cold exposure [13]. The FGFs are cell-signaling proteins with diverse functions that include the regulation of metabolism and cell development and repair [56]. The human FGF family contains 22 members, of which FGF19, FGF21, and FGF23 belong to the endocrine (hormone-like) subfamily, that regulate cell metabolic activities and function through the protein family, klotho [56]. The main cofactors, α-klotho and β-klotho, act to tether FGF 19/21/23 to their receptor(s) forming an active ternary complex [56]. Human-based studies reveal increased circulating levels of FGF21 in MAFLD, through activation of Peroxisome Proliferator-activated Receptor α (PPARα) [13,57]. FGF21 enhances VLDL disposal in adipose tissue, reduces hepatic VLDL secretion, and promotes pancreatic β-cell function and insulin secretion [13,58]. As such, FGF21 represents a potential therapeutic agent for the management of T2D and metabolic syndrome (although FGF21 analogues do not appear to lower blood glucose in human-based studies) [13]. In addition to its peripheral effects on lipid metabolism, FGF21 may also influence central hypothalamic appetite control for sweet foods (simple sugars, but not complex carbohydrates, lipids, or proteins) [59,60]. Data from a wide range of settings (including rodent, primate, and human-based data) show that FGF21 may reduce the consumption of sweet foods [59]. In this way, the liver may regulate macronutrient-specific intake through the release of FGF21, acting as a satiety signal centrally to suppress the intake of sweet food [60]. This liver-to-brain hormonal axis is a negative feedback loop, with sucrose ingestion resulting in the elevation of hepatic FGF21 production [60].

Retinol binding protein 4 (RBP4) transports vitamin A as retinol, and is secreted by the liver and adipose tissue [13]. Rodent-based studies suggest an effect of RBP4 on inflammation and IR in adipose tissue [13,61]. However, in another study in which RBP4 was overexpressed within the liver, there was no appreciable effect on glycaemic control despite increased serum levels of RBP4 [13,62]. Regarding serum levels of RBP4 in MAFLD, existing data are conflicted. Some studies show that serum levels of RBP4 are increased in the context of MAFLD, T2D, and IR [13,63]. However, in a comparison of RBP4 levels between participants with chronic kidney disease (CKD), chronic liver disease (CLD), and controls, it was shown that compared with controls, RBP4 levels were highly increased in CKD but equivalent in CLD [64]. Furthermore, in hepatic dysfunction, RBP4 levels were decreased although the relative concentrations of isoforms were not affected [64]. It has been suggested that impaired catabolism of RBP4 in the kidneys (in the context of CKD) may result in an accumulation of RBP4 isoforms in the serum [64,65]. Based on the data outlined, it is unclear whether MAFLD may associate with serum levels of RBP4 and the nature of any such association. Furthermore, any potential role for RBP4 in mediating peripheral IR and other metabolic effects in MAFLD remains incompletely understood [13]. Proposed mechanisms include the activation of adipose tissue macrophages and the direct activation of hepatocellular lipogenic programs [35].

Sex hormone-binding globulin (SHBG) is a transporter of sex steroids [13]. Hepatic synthesis of SHBG is upregulated by estrogen and inhibited by testosterone [66]. Therefore, serum levels of SHBG are influenced by sex (pre-menopausal women typically have higher levels than men), and through the use of the combined oral contraceptive pill (in which the ethinylestradiol component is a potent stimulator of hepatic SHBG production) [66]. Hepatic steatosis is associated with lower serum levels of SHBG (likely mediated through the effects of obesity), with a negative correlation between serum SHBG levels and IR [13]. In the context of IR and hyperinsulinaemia there is lower expression of hepatocyte nuclear factor 4α, a key transcription factor of SHBG [13]. Furthermore, inflammation in MAFLD may precede reduced hepatic expression of SHBG [13]. Overexpression of SHBG suppresses lipogenesis and thereby can diminish the development of hepatic steatosis [13,67]. Therefore, the suppression of hepatic SHBG release in the context of MAFLD may play a role in the causative pathway for hepatic steatosis in this condition, and contribute towards worsening of hepatic IR. Furthermore, suppressed levels of SHBG may also act as a useful clinical biomarker for IR (including for other IR-associated conditions such as polycystic ovary syndrome [PCOS] [68] and more generally such as the young male population [69]). Furthermore, the detection of a low serum level of SHBG should act as a prompt to further investigate and manage the possible development of MAFLD.

### 3.2. Liver-Secreted Metabolites

The major liver-derived metabolite classes include ketones, lipoproteins, bile acids (BAs), and acylcarnitines [13]. In this sub-section, we discuss the role of each of these metabolites as a mediator of peripheral IR and dysmetabolic milieu in the context of MAFLD.

Ketones: Ketone bodies that are synthesized by the liver include acetone, acetoacetate, and β-hydroxybutyrate [13]. Within peripheral tissues, ketone bodies are converted back to acetyl-CoA as a substrate for energy production [13]. Ketone bodies also regulate metabolism within the peripheral tissues and central nervous system, including the hypothalamic regulation of appetite, with improvements in systemic insulin sensitivity and reduced adiposity [13]. Despite the known effects of ketone bodies on central appetite regulation, including acute appetite suppression following the consumption of a ketone ester drink [70], there is little evidence to support the weight-losing effects of keto supplements. One recent systematic review on the effects of 7-keto-DHEA on weight loss was inconclusive, with a recommendation by the authors that further studies are required to clarify the efficacy and safety of 7-keto-DHEA [71].

Beyond central appetite regulation, β-hydroxybutyrate also regulates lipid metabolism within adipocytes and reduces oxidative stress through the inhibition of class 1 histone deacetylase (HDAC), which, in turn, contributes towards improved insulin sensitivity [13,72,73]. Subsequent activation of PPARα and increased expression of genes implicated in lipid metabolism results in increased oxidation of hepatic fatty acid, plasma clearance of triglycerides, and FGF21 production [13,74,75]. Regarding liver-secreted acetoacetate, there may be a role in the amelioration of diet-induced hepatic fibrosis [13,76]. Regarding liver-secreted acetone, although it is taken up by tissues, its effects on insulin sensitivity and glycaemic control are unknown [13]. Unfortunately, despite the insights outlined above, there is conflicting evidence in the current literature regarding the hepatic ketogenic response to MAFLD, and the potential impact of such liver-derived ketones on peripheral insulin sensitivity and glycaemic control [13]. Some of this uncertainty may stem from differences in trial design and variable responses to ketogenic dietary interventions within the population [13]. A challenge for the future will be to clarify the hepatic ketogenic response to MAFLD and its potential impact on peripheral insulin sensitivity.

Uric acid: In humans, purine catabolism results in the production of the oxidation product, uric acid [77]. Uric acid is produced in the liver, but also in muscle and adipose tissue, and excreted in the urine [77]. Hyperuricaemia associates with features of the metabolic syndrome that are unified by abdominal obesity and IR [77]. Rodent-based studies suggest that high uric acid levels may regulate oxidative stress, inflammation, and enzymes that control lipid and glucose metabolism [77]. Furthermore, humans lacking uricase (the enzyme that breaks down uric acid) develop metabolic dysfunction [77]. The role of uric acid in the development of MAFLD remains unclear and should be a focus of future research.

Lipoproteins: Hepatic secretion of lipids occurs primarily through VLDL, consisting of triglycerides (55%), but also cholesterol and cholesterol esters (25%) and phospholipids (20%) [13]. Hyperglycaemia contributes to the hepatic secretion of lipids. When delivered to the liver in large quantities, glucose is converted to glycogen and then enters the glycolysis pathway resulting in de novo lipogenesis [30]. Following the storage of lipids as intra-hepatic triglycerides, these are exported from the liver as VLDL [30]. Liver X receptors (LXRs, members of the nuclear receptor family) are implicated in the regulation of intra-hepatic lipid biosynthesis, glucose homeostasis, and cholesterol metabolism [30]. In hepatocytes, LXRα expression is induced by insulin and oxysterols, with increased expression of lipogenic enzymes and suppression of key gluconeogenic enzymes [30].

MAFLD is associated with increased secretion of VLDLs and a reduced capacity for insulin-mediated suppression of VLDL secretion from the liver [13]. Cleavage of fatty acids from triglyceride within VLDLs occurs through the action of lipoprotein lipase on the surface of endothelial cells in capillaries [13]. Increased availability of fatty acids from the hepatic secretion of VLDL can cause peripheral IR, and is associated with obesity and hepatic steatosis [13]. Following the depletion of triglyceride, lipoprotein particles form intermediate-density lipoproteins or low-density lipoproteins (LDLs). The accumulation of cholesterol within the plasma membrane of skeletal muscle associates with IR secondary to a reduction of Glucose Transporter 4 (GLUT4) insertion within the membrane [13,78]. Furthermore, transfer to the plasma membrane of skeletal muscle may also contribute towards peripheral IR through reduced insulin-stimulated GLUT4 translocation and inflammation in macrophages [13,79]. Finally, liver-derived ceramides (dependent on the supply of fatty acids [80]), also contribute towards the dysregulation of adipose tissue lipolysis [13].

BAs: Within the liver, BAs (deoxycholic acid, cholic acid, and chenodeoxycholic acid) are formed by the oxidation of cholesterol [13]. BAs play an important role in solubilizing lipophilic nutrients in the small intestine [81]. Most studies show a positive correlation between circulating and liver BAs with progressive MAFLD [13,82]. BAs induce favourable metabolic effects (unlikely to be mediated directly within the muscle and adipose tissue), including improved insulin sensitivity and glucose metabolism through activating the G protein-coupled receptor, ‘Takeda G protein–coupled receptor-5′ (TGR5) [13,83]. BAs also activate the Farnesoid X receptor (FXR, a nuclear receptor) [13] in ileal enterocytes, and thereby induce the expression of fibroblast growth factor (FGF) 15/19 which functions as a hormone [81]. FGF 15/19 then acts on a cell surface receptor complex in hepatocytes to suppress gluconeogenesis and BA synthesis and stimulate glycogen and protein synthesis and gallbladder filling [81]. In this way, the BA-FXR-FGF 15/19 signaling pathway regulates the enterohepatic response in the post-prandial period [81].

Acylcarnitines: The liver is the main source of acylcarnitine, which is an important energy source, particularly in the fasting state [13]. Acylcarnitine (formed from the coupling of acyl-CoA and carnitine) is transported into mitochondria, where regenerated acyl-CoA is then oxidized through β-oxidation [13]. Although it is not clear whether circulating acylcarnitines impair systemic insulin action and glucose homeostasis, they are taken up by pancreatic β-cells (leading to insulin depletion) [13]. Acylcarnitines are also a likely fuel source for thermogenesis within brown adipose tissue during cold exposure and can activate pro-inflammatory signaling in macrophages [13,84]. Based on the current literature, it is not possible to formulate a clear association between MAFLD and plasma acylcarnitines [13]. Plasma levels of acylcarnitines vary according to the stage of MAFLD, with reduced levels in hepatic steatosis [85] and increased levels in NASH [13,86]. These changes could stem from variations in hepatic mitochondrial β-oxidation during the progression from MAFLD into NASH [13].

### 3.3. Liver-Derived miRNAs

Many miRNAs (including those that regulate insulin secretion and sensitivity) are increased both in the liver and circulation in the context of MAFLD [13,87]. Examples include miR-802 [88] and miR-144 [89] which impair insulin sensitivity within both the liver and periphery (adipose tissue and skeletal muscle) [13]. It is beyond the scope of this review to provide a detailed account of miRNAs in the context of MAFLD, provided in detail elsewhere [90]. Briefly, several miRNAs (including miR-122, miR-33, miR-34a, and miR-21) likely regulate hepatic lipid metabolism, and hold therapeutic potential in MAFLD [90]. Furthermore, other miRNAs (including miR-122, miR-20-5p, and miR-29) may modulate hepatic glycogen metabolism, and potentially divert glucose towards de novo lipogenesis [90]. Certain miRNAs (such as miR-21) may also play a role that extends beyond the pathogenesis of MAFLD to the promotion of tumorigenesis [90]. Finally, certain miRNAs (such as miR-122 and miR-34a) may play a future role as biomarkers for the diagnosis of MAFLD, and to indicate the stage of disease (including the discrimination between MAFLD and NASH) [90].

Currently, we lack causal data to prove that liver-derived miRNAs are implicated in regulating insulin sensitivity in MAFLD, given that such miRNAs are also expressed and secreted from other tissues [13]. It is important to note that miRNAs represent a relatively small fraction of all non-coding RNAs, which have the capacity to regulate gene expression [13]. A challenge for the future is to elucidate the role of miRNAs and other non-coding RNAs in the regulation of insulin sensitivity and the pathophysiology of peripheral IR in the context of MAFLD.

## 4. Gut-Related Factors in the Development of MAFLD

Having focused on the complex interlinks between MAFLD and IR within the periphery, it is also important to consider the mechanisms by which gut-related factors interact with the liver and contribute towards the development of MAFLD. It is beyond the scope of this concise review to provide an exhaustive overview of this topic, which has been provided elsewhere [91]. Rather, we provide an outline of some key insights.

Compared with healthy controls, MAFLD is associated with lower levels of *Bacteroidetes* and higher levels of *Prevotella* and *Porphyromonas* species [10,92]. The interaction of these gut microbiota and other gut-derived products with the liver, and their role in the development of MAFLD is influenced by the integrity of the gut mucosal wall [93]. The integrity of the gut mucosal wall is, in turn, influenced by the gut microbiota and multiple other factors that include diet and lifestyle, alcohol intake, and stress. Broadly, the gut interacts with the liver via the portal vein. Beyond hepatic steatosis, the inflammatory elements of MAFLD are mediated via the expression within the host of toll-like receptors (TLR) 4 or 9, and TNF-α receptor. These inflammatory elements can be stimulated by translocated gut microbiota and/or microbial metabolites that include secondary BAs and trimethylamine [10,92]. Short-chain fatty acids (SCFAs, including propionate and acetate) are microbial metabolites that are produced within the gut through the fermentation of dietary fibre by the gut microbiota [94]. SCFAs may alter metabolic processes (including lipid storage) within the liver either as energy substrates (thereby contributing towards the development of MAFLD) or mediating anti-inflammatory effects through stimulation of the free fatty acid receptor (FFAR) 1 and 2 (thereby having a protective effect on the liver) [94]. The relative production of propionate versus acetate by the gut microbiota may influence their cumulative effect on MAFLD development [94].

Secondary BAs: BAs facilitate the intestinal absorption of dietary lipids and fat-soluble vitamins [95,96]. Primary BAs (accounting for 70–80% of the total BA pool) are synthesised from cholesterol within the liver. Secondary BAs are microbial metabolites that are derived from primary BAs through their modification by the gut microbiota [95]. Most BAs are recycled via the enterohepatic circulation, with only around 5% of BAs being lost in the faeces [95,97]. As a whole, BAs have variable chemical properties regarding hydrophobicity and conjugation status [95]. Such chemical variability of BAs underlies the heterogeneity of their functional propensity for nuclear receptor binding, nutrient digestion, and the maintenance of intestinal mucosal integrity [95]. These functional properties of BAs play a key role in mediating complex interactions between the gut and the liver.

The TGR5 is a key nuclear receptor target of BAs [95]. In a murine model, it was demonstrated that BA-induced stimulation of TGR5 receptors resulted in the release of GLP1 from intestinal L-cells [98,99]. Through its incretin effect of enhanced post-prandial insulin release from the islet ß-cells, enhanced GLP1 release improves overall metabolic health (including glucose, lipid, and cholesterol metabolism). This provides a mechanism whereby BAs (through their modification by the gut microbiota) can impact metabolic liver health and influence the aetiology of MAFLD [98,99]. The FXR (introduced above) is a further nuclear receptor target of BAs that regulates numerous downstream signaling cascades. These include the PPARs that influence insulin sensitivity [95]. Through their actions on nuclear receptors such as TGR5 and FXR that, in turn, influence important metabolic pathways, and through their effects on maintaining intestinal mucosal integrity, BAs mediate the gut–liver axis and play a key role in maintaining the metabolic functioning of the liver. In the context of dysbiosis, or other environmental insults (such as alcohol excess), dysregulated BA signaling (through aberrant processing of primary BAs) likely contributes towards metabolic dysfunction, including the development of MAFLD through inflammatory and fibrotic processes [95].

Trimethylamine: Through the metabolism of methylamine-containing nutrients (lecithin, L-carnitine, and choline), the gut microbiota produces trimethylamine (TMA), a microbial metabolite [100]. The main sources of TMA are animal products, for some of which an association between high intake and cardiometabolic risk has been shown (red meat, but not fish) [101,102,103]. TMA is processed to trimethylamine N-oxide (TMAO) via the action of hepatic flavin monooxygenases (FMO), following its delivery to the liver via the portal vein [100]. FMO3 (regulated via the FXR receptor through the action of BAs) is the primary enzyme that converts TMA to TMAO [100]. In humans, plasma levels of TMAO correlate with cardiovascular risk [100,104]. Furthermore, in a rodent model, *Fmo3* knock-down resulted in a reduction of atherosclerotic lesions [100,105]. However, in other rodent-based studies, the data are inconsistent and contentious regarding the impact of TMA on the size of aortic atherosclerotic lesions [100]. Future studies should explore further the potential pathophysiological role of TMA and TMAO in the development of MAFLD.

A detailed overview of the complex mechanisms that link intestinal dysbiosis with the development of hepatic inflammation and fibrosis is beyond the scope of this review and has been covered elsewhere [106]. Briefly, the origin of MAFLD appears to be the accumulation of hepatic lipids. The gut and liver communicate with each other in a bi-directional way via the biliary tract, portal vein, and systemic circulation [106]. Gut microbial metabolites (outlined above) and microbial-associated molecular patterns, can directly modulate lipid metabolism and the inflammatory response to lipid accumulation within the liver, thereby influencing the pathogenesis of MAFLD [106]. Liver components such as BAs and immunoglobulin A influence intestinal health (including intactness of the gut wall), completing the bi-directional communication between the gut and the liver [106]. Over time, this process can result in hepatic fibrosis and cirrhosis.

## 5. Concluding Remarks: How to Optimise Metabolic Liver Health

Despite the prominent role of the liver in determining overall metabolic health, MAFLD suffers as a clinical entity for three main reasons:

(i) The clinical development of MAFLD is usually insidious, and its onset is often asymptomatic, with important pathogenic factors including progressive weight gain in the context of an underlying genetic predisposition;

(ii) We lack a cheap, accurate, and non-invasive diagnostic test for MAFLD. Although MR spectroscopy provides images that would facilitate a diagnosis of MAFLD, the expense of this technique precludes its widespread clinical adoption. Currently, liver biopsy is the only procedure that can accurately and reliably diagnose MAFLD. Unfortunately, the general applicability of liver biopsy is precluded by its invasiveness. The liver function test (LFT) as a general screener for liver disease is notoriously unreliable for diagnostic accuracy and sensitivity [107]. Furthermore, liver fat indices may fail to reflect treatment-induced changes in liver fat content [108,109];

(iii) Currently, MAFLD lacks bespoke therapies that have been designed, assessed, and approved specifically for the purpose of managing this condition. The dawn of the dual and triple incretin agonist therapies (including those that combine GLP1 with glucagon such as cotadutide with associated improvements in hepatic fatty index), will likely feature prominently in future treatment algorithms for MAFLD [110].

Accordingly, there is a disconnection between the metabolic health implications of MAFLD and our limited clinical armamentarium with which to accurately diagnose and effectively manage this important condition. The development and implementation of specific pharmacotherapies for MAFLD, including the dual and triple incretin agonists [110] will help to raise awareness of MAFLD, and hopefully catalyze the development of accurate and reliable non-invasive diagnostic tests. Such novel pharmacotherapies for MAFLD should also provide an impetus for both patients and healthcare professionals to engage in timely screening for MAFLD, enabling prompt initiation of an effective management strategy. Such a scenario will surely raise the profile and awareness of MAFLD amongst the public, media, and healthcare professionals.

Future therapeutic strategies should also focus beyond reversing the pathogenesis of MAFLD, towards improving peripheral IR. Thiazolidinediones (TZDs) are established therapies for T2D that enhance insulin sensitivity in adipose tissue, skeletal muscle, and liver, and promote the re-distribution of fat from the liver and skeletal muscle to adipocytes. TZDs mediate their effects through the activation of PPARγ that, in turn, regulates several genes implicated in the metabolism of glucose and lipids [111]. Metformin may also improve peripheral insulin sensitivity through the reduction of hepatic glucose production [111]. A future therapeutic strategy for MAFLD-related peripheral IR is the enhancement of muscle mass and fat oxidation within skeletal muscle. The skeletal muscle accounts for 70–80% of total glucose disposal after insulin stimulation [111]. In a rodent-based study, the over-expression of muscle-related uncoupling protein 3 (UCP-3, which dissipates mitochondrial proton gradients) was shown to protect against muscle-related IR in response to a high-fat diet [111,112]. Somewhat unexpectedly, deletion of the autophagy-related gene-7 (*ATG-7*, specific to skeletal muscle) in mice conferred protection from diet-induced obesity and IR and was accompanied by increased fat oxidation and adipocyte browning [111,113]. The authors proposed possible mediation of this pathway via the promotion of Fgf21 expression, stemming from deficiency of autophagy with subsequent mitochondrial dysfunction [111,113]. Other therapeutic targets to improve skeletal muscle insulin sensitivity include PPARδ and myostatin [111].

In addition to novel pharmacotherapies targeted at MAFLD and IR, a key strategy to improve the metabolic liver health of the population should involve the encouragement and facilitation of a high-fibre diet [16]. In addition to the optimised intake of dietary fibre, sustained weight loss represents an extremely effective therapy for MAFLD, with the demonstration of reversal of hepatic steatosis and improved hepatic insulin sensitivity and glycaemic control following moderate weight reduction of T2D [114,115]. Future studies should further explore the effects of dietary glycaemic indices on hepatic insulin sensitivity, including the pharmacological potential impact of regulating the action of key incretin hormones (such as GIP) on insulin sensitivity, with insights from rodent-based studies [116,117,118].

Finally, MAFLD has transitioned to global prominence in recent years and now affects one in four of the world’s population. MAFLD has surpassed alcohol excess as the commonest cause of chronic liver disease and is an important contributor to the development of cirrhosis [9,10]. The adverse health implications of this MAFLD behemoth will dominate healthcare systems globally like a tsunami for decades to come. We all need to wake up, smell the coffee, and address the elephant in the room. The elephant is MAFLD. The coffee is OUR call to act now.

## Figures and Tables

**Figure 1 metabolites-13-00757-f001:**
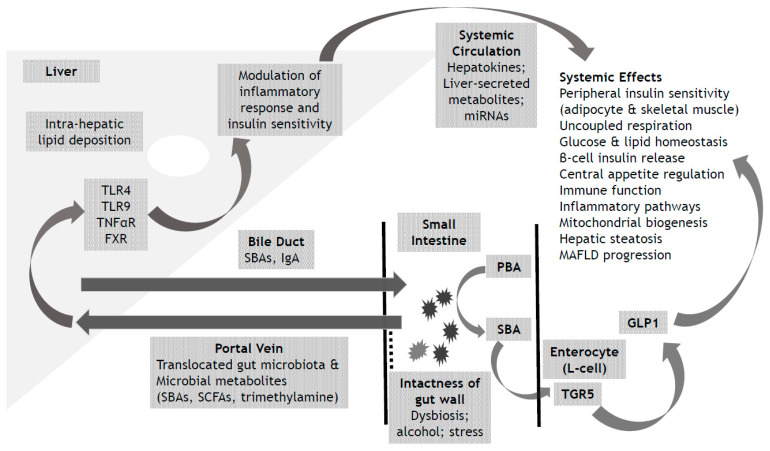
Schematic outline of interplay between gut related factors, MAFLD and peripheral Insulin Resistance. FXR: Farnesoid X Receptor; GLP1: Glucagon like Peptide 1; IgA: Immunoglobulin A; MAFLD: Metabolic-associated Fatty Liver Disease; miRNA: micro RNA; PBA: Primary Bile Acid; SBA: Secondary Bile Acid; SCFA: Short Chain Fatty Acid; TGR5: Takeda G protein coupled receptor 5; TLR4: Toll like Receptor 4; TLR9: Toll like Receptor 9; TNFα R: Tumour Necrosis Factor α Receptor.

**Table 1 metabolites-13-00757-t001:** Mechanisms linking MAFLD with peripheral Insulin Resistance.

Sub-Group	Liver-Derived Factor	Regulatory Mechanisms
Hepatokines	Activin E Fetuin A Fetuin B Visfatin Chemerin DPP4 FGF21 RBP4 SHBG	Uncoupled respiration Peripheral insulin sensitivity Insulin receptor pathway Cellular metabolism and appetite control Glucose and lipid homeostasis Insulin release from β-cells Mitochondrial biogenesis Immunity Hepatic steatosis
Liver-secreted metabolites	Ketones Uric Acid Lipoproteins Bile Acids Acylcarnitines	Lipid metabolism Peripheral insulin sensitivity Appetite control Oxidative stress and inflammation Intra-hepatic lipid synthesis Fuel source for thermogenesis
Liver-derived mi-RNAs	miR-802 miR-144 miR-122 miR-33 miR-34a miR-21	Peripheral insulin sensitivity Hepatic lipid metabolism Hepatic glycogen metabolism Promotion of tumorigenesis (Future biomarkers of MAFLD)

DPP4: dipeptidyl peptidase-4; FGF21: fibroblast growth factor 21; MAFLD: metabolic-associated fatty liver disease; mi-RNA: micro RNA; RBP4: retinol-binding protein 4; SHBG: sex hormone-binding globulin.

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
