# Peer review of "Metabolic-Associated Fatty Liver Disease and Insulin Resistance: A Review of Complex Interlinks"

_metabolites, 2023, doi:10.3390/metabo13060757_

Round 1

Reviewer 1 Report

The manuscript by Barber et al aims to provide an overview of the complex  mechanisms whereby gut-related factors influence the development of MAFLD, and MAFLD influences the development of insulin resistance within the periphery.  However, in my opinion these aims were not achieved due to the fact that some crucial informations were omitted including, deatiled relationships between insulin signaling pathway, inflammation, metabolism of carbohydrate and fats  in insulin sensitive tissues and  pathways related with inflammation and fibrosis of the liver. The manuscript should be improved according to the following comments:

  1. The chapter 2 describing the relationship between MAFLD and insulin resistance presents these two disorders as a vicious cycle. Considering  indicated by authors hyperinsulinemia state in insulin resistance, one may notice that beta cells of pancreas  are able to produce insulin. However, the authors did not describe insulin resistance state in type 2 diabetes, in which insulin production and secrection is significantly attenuated. This is a serious shortcoming as it is estimated that  25 - 70% of MAFLD patients suffer from type 2 diabetes.

 2. NAFLD is used instead of MAFLD. Why did authors use MAFLD term? What is the difference between these two terms?

3. Generally, in chapter 3 entitled "Effects of MAFLD on peripheral IR" only for few hepatokines the mchanisms leading to peripheral IR were presented, including fetuin A, DPP4,FGF21, SHBG. The rest of hepatokines were described superfacially.  Peripheral IR concerns mainly liver, skeletal muscle and adipose tissue. Taking the liver as a  key organ in the development of MAFLD, the relations betwenn hepatokines and biomolecules on the level of signaling pathways in skleletal and adipose tisse should be presented 

4. Chapter 3.4 is poor. Numerous papers presents microRNA involved in/ associated with MAFLD, as an example please see http://dx.doi.org/10.1136/gutjnl-2018-318146

5. Chaper "Gut-related factors in the development of MAFLD" lakcs details how dysbiosis induces formation of inflammation process and leads to liver fibrosis.

6. The relationships between hepatokines, hepatic metabolites and crucial pathways leading to IR in the adipose tissue, skleletal muscle and liver sholud be presented/ summarized in the figure. 

7. Similarly,  the connections between gut dysbiosis, inflammation and development of MAFLD should also be showed on the scheme. 

8. Conclusions are too long. The majority of presented informations should be moved from Conclusions to the Introduction (first six paragraphs) to show the justification for the topic of the manuscript.

Minor editing of English language required

Reviewer 2 Report

The work discusses in detail the relationship between MAFLD and IR, which is often positive feedback. Due to the high level of complexity of these dependencies, I propose to illustrate the text with a figure showing the most important phenomena.

In addition, I recommend extending the work to discuss possible methods of IR treatment and possibly indicate why they have such low effectiveness.

Reviewer 3 Report

Dear  Authors,

I believe that the article submitted for evaluation is well written and concise. I have no comments as to the content, I only think that the value of the work would be increased by the scheme of interactions between the intestines and the liver and their impact on IR.

Reviewer 4 Report

This is a comprehensive review of the interlinks between Insulin resistance (IR) and metabolic fatty liver disease (MAFLD). The paper is well organized and presented. However there are certain points that should be attended by the authors.  A comprehensive table indicating the various factors implicated in MAFLD and IR is important for the reader to better understand this subject. Moreover a figure should be included to demonstrate how these factors inluence IR. In general most of the information provided by this paper is an abbreviated version of two previous reviews (Ref 13 and 34) . The new information is chapter 4 on Gut related factors.

Minor point: Ref 27  is referring to Activin A and not to activin E as suggested in the text Line 137)

Round 2

Reviewer 1 Report

The authors responded sufficiently to the majority of my comments. I recommend to accept manuscript inthe current form.

Reviewer 4 Report

The authors have satisfied the recommendations made by the reviewers.